# Interconnections between Content Knowledge and Pedagogical Content Knowledge of a University Lecturer in Linear Algebra

**Diana Vasco-Mora** [1] , **Nuria Climent-Rodríguez** [2,*] **and Dinazar Escudero-Ávila** [3]

1   Faculty of Livestock Sciences, Universidad Técnica Estatal de Quevedo, Quevedo 120501, Ecuador; dvasco@uteq.edu.ec
2   Faculty of Education, Psychology and Sports Sciences, Universidad de Huelva, 21007 Huelva, Spain
3   Independent Researcher, 28035 Madrid, Spain; eadinazar@gmail.com
*   Correspondence: climent@uhu.es; Tel.: +34-959-219-261 or +34-664-661-603

**Abstract:** The aim of this study is to deepen our understanding of the practice of a lecturer in linear algebra by exploring the connections he makes between his content knowledge and his pedagogical content knowledge while working on the topic of matrices. Data were collected through video recordings of his classes and semi-structured interviews, and were analysed with the Mathematics Teacher's Specialised Knowledge model. Instances of classroom performance, supported by the teacher's own affirmations, provided evidence relating to the categories comprising the model, and enabled us to establish connections between the lecturer's knowledge, his understanding of his students' learning capabilities, and his knowledge of teaching mathematics, which together account for his classroom practice: the use of varied examples to introduce new content, the highlighting of the most salient aspects of the topic, and alerts about potential errors and difficulties. The contribution that these results could make to the training of university teachers, which would be done with the knowledge of the areas of difficulty shown by the teacher in mind, could be used to deepen other elements of their pedagogical content knowledge. The interconnections between areas of knowledge identified by the study also serve to validate the usefulness of a theoretical model for studying teachers' knowledge.

**Keywords:** case study; linear algebra; matrices; teacher knowledge; tertiary education; learning theory; content analysis

## 1. Introduction

Over the last decade attempts have been made to understand the ways in which teachers' knowledge supports their practice. Surprisingly, there have been few studies (insofar as mathematics teaching is concerned) into nursery- and university-level education [1].

It might seem that in the case of nursery teachers pedagogical content knowledge [2] is needed more than anything else (certainly more than content knowledge), whilst in the case of university mathematics lecturers the reverse is true, and content knowledge is unquestionably central, with less consensus on the relative importance of the pedagogical knowledge relating to this content. Nevertheless, research into university lecturers' knowledge has emphasised the need for both components of knowledge [3], and studying mathematics lecturers' knowledge can raise interesting questions around the kind of teacher training they should receive.

In this paper, we explore the interrelationships between the different kinds of knowledge brought into play by a university lecturer in the course of his practice. Specifically, we raise the question of what elements of content knowledge and pedagogical content knowledge relating to linear algebra underpin this practice, and how the two are interconnected. In order to formulate our answer, from the perspective of the Mathematics Teacher's Specialised Knowledge model [4], we focus on one aspect of content knowledge

(denominated as Knowledge of Topics in the model) and on two facets of pedagogical content knowledge relating to the teaching and learning of this content.

This study takes an interpretative focus [5], that is, more than aiming to measure the lecturer's knowledge, identifying the presence or absence of essential elements required for teaching mathematics, we are interested in understanding the knowledge which the lecturer brings into play in the classroom, and the interrelationships between the various components which comprise it. Our interest in this field stems from the fact that there have been various studies into the teaching and learning of linear algebra at university level [6], but these have been predominantly oriented towards the difficulties faced by students [7]; very few have focused on the knowledge deployed by lecturers in linear algebra [8].

We hope that studies such as this will contribute in the future to the design of grounded training programmes for university lecturers. In this regard, the instances revealed in this study of how a teacher's knowledge interacts with their practice, and the interconnections identified between aspects of this knowledge, are valuable for the ideas they give for developing the different components of teachers' knowledge.

## 2. Theoretical Framework

The interest of this study focuses on teacher knowledge, understood as a resource to be drawn on in the everyday running of classroom events [9]. This section is divided into two parts, the first dealing with the background to studying mathematics teacher knowledge at both secondary and university levels, and the second describing the theoretical model applied to this study.

### 2.1. Background

Various theoretical frameworks have been developed specifically for the study of teacher knowledge, many of which are founded on the categories originally proposed by Shulman [2,10] for categorising what a teacher needs to know to carry out their work effectively. In particular, the majority of models following in the wake of Shulman featured the domains of content knowledge and pedagogical content knowledge—that is, knowledge specific to enabling the teaching and learning of content.

There has also been an upsurge of interest in teachers' knowledge regarding the teaching of algebra. Artigue [6], studying the algebraic knowledge of prospective secondary teachers, considered three dimensions; the epistemological, the cognitive, and the didactic (referring to knowledge of algebra and its historical development, learning algebra, and teaching algebra, respectively). Within the epistemological dimension a distinction is made between algebra as a tool (for solving problems, whether intrinsic or extrinsic to mathematics) and as an entity in itself (that is, a structured set of objects, each with its own properties, modes of representation, and means of treatment, such as functions and structures similar to vector spaces). The study found that the prospective teachers regarded algebra as a domain restricted to the field of algebraic theory (the epistemological dimension), in which algebraic techniques were limited to the syntactic part as a set of rules. As a result of their teaching practice, the prospective teachers developed greater sensitivity to the difficulties their students faced with the symbolic system and with the conversion from one semiotic register to another (cognitive dimension). The paper concluded that a good understanding of linear algebra means having the cognitive flexibility to move between different semiotic registers and languages, such as the geometric, the algebraic, and the abstract (of vectors and linear transformations).

With respect to teachers' knowledge of linear algebra (specifically matrices), Sosa [11,12] describe the pedagogical content knowledge of two final-year secondary teachers in terms of the features of learning mathematics and the potential and use of examples (knowledge of teaching). The description is carried out by means of indicators of teacher knowledge regarding the language used by students to talk about content, student needs and difficulties, misunderstandings arising from generalisations from previous content, erroneous ideas, and errors resulting from disregarding mathematical conventions.

Studies into the teaching of mathematics at the university level have also been carried out, covering the questions of task design, teaching strategies, and interaction with students [13], and these have also included consideration of linear algebra, such as the case study of university teaching by Jaworski [14]. This analysis illustrates teacher knowledge of the main difficulties faced by students and identifies two modes of discourse: the expository mode (the lecturer focuses strictly on the mathematical features of the topic) and the didactic mode (the lecturer focuses on constructing student understanding of the material).

In an analysis of the knowledge displayed by an experienced university lecturer giving a course in differential equations, Speer [15] drew attention to a generative relationship between the instructor's specialised content knowledge and his pedagogical content knowledge. They suggest that the former can support teachers' work by enabling them to learn from their practice, the result of which has the potential to be incorporated into their stock of pedagogical content knowledge.

Another essential element of teacher knowledge is the use of examples in teaching. The teacher's deployment of examples reveals facets of their knowledge for teaching mathematics. This includes learner-oriented attributes such as transparency, whereby the salient features of the chosen example highlight the target concept or procedure, and generalisability, by which the necessary characteristics of what is being exemplified are contrasted with what is arbitrary and changeable [16]. In Mali [17], the use of generic examples was found to be characteristic of the tutoring style of an experienced university lecturer in linear algebra. They found that an important feature of a generic example was to reveal the properties of mathematical concepts, and that the use of such examples mediates student understanding and is indicative of teaching style.

Elsewhere, Figueiredo [18] provided a typology of examples according to the following categories: definition (those placed immediately after a definition); representation (those providing initial contact with a concept); characteristics (those occurring after discussion of a concept and intended to highlight essential features); internal applications; and external applications. With respect to the category of characteristics, the authors note that these depend largely on the teacher's foresight, experience, and originality, especially with regard to alerting students to aspects of potential difficulty.

The dimensions of possible variation can be applied to the use of examples to encourage students to understand which elements of mathematical content can undergo variation [19]. In variation theory, the teacher selects activities which focus on what varies and what stays the same when small changes are introduced into an example, thus drawing students' attention to the fundamental attributes of the topic [20]. Examples can be considered according to contrast (in order to identify which elements of a mathematical object are the same and which are different); generalisation (in order to recognise the defining features of a mathematical object in different contexts); and fusion (in order to see which features change and which stay the same) [21].

In conjunction with the use of examples in teaching, we can also mention the use of error as an important part of the teacher's knowledge. In this regard, González [22] identifies the use of error by 26 secondary mathematics teachers and argues that one such use is to improve the learning process. This is achieved when the teacher is able to analyse potential errors which the students might make and anticipate these through leading questions so as to make them aware of the error.

The studies mentioned above share an interest in exploring the relationship between practice and knowledge in the context of university teaching. Nevertheless, it is still the case that there are few studies of lecturers' knowledge at this level, and there remains a need to carry out research into the knowledge of linear algebra teachers, as pointed out by Fukawa-Connelly [23].

### 2.2. The Mathematics Teacher's Specialised Knowledge Model

The last three decades have seen an expansion of research into mathematics teachers' knowledge, with various theoretical models proposed as a framework for analysis [24].

In this study, we use the Mathematics Teacher's Specialised Knowledge framework (henceforth MTSK) [25]. As with many other models, the MTSK model is founded on the categories proposed by Shulman and arose from the necessity to improve the characterisation and delimitation of the subdomains common content knowledge and specialised content knowledge raised in mathematics teaching in the Mathematical Knowledge for Teaching model (MKT) [26], and to remedy the difficulties encountered in applying the MKT model to the research we had carried out with teachers, difficulties which were also reported by Silverman [27].

The MTSK model is intended to be used in the analysis of teachers' practices by linking professional actions and decisions to the knowledge underpinning them. The aim of the model is to capture the specificity of this knowledge, and hence focuses only on knowledge, which in its entirety is uniquely meaningful for mathematics teachers. In contrast to the MKT model, the MTSK model considers all the subdomains comprising the model as specialised and is intended as a useful framework for considering many types of specialised knowledge required by both mathematics teachers and teacher educators [28].

The model recognises three broad domains: Mathematical Knowledge, Pedagogical Content Knowledge and Beliefs (the latter constituting a component permeating the other two domains). In turn, each domain is divided into subdomains (see Figure 1), for each of which a set of categories has been developed for a more fine-grained analysis.

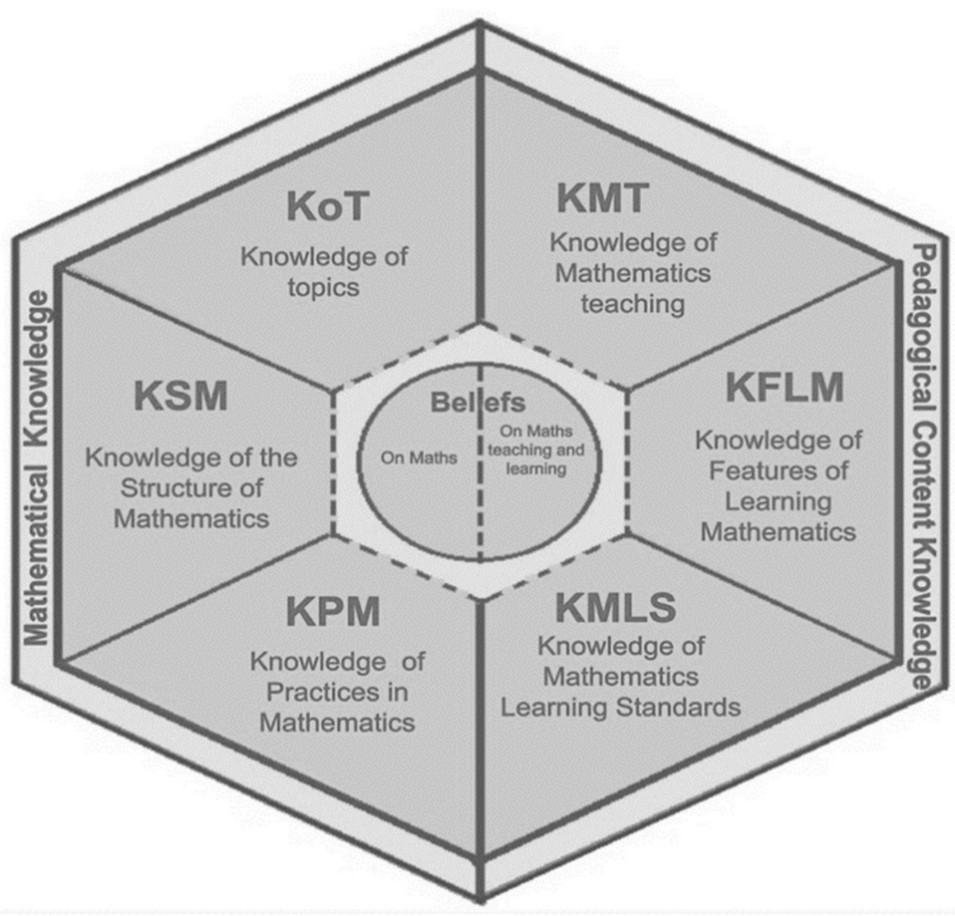

**Figure 1.** Domains and subdomains of the Mathematics Teacher's Specialised Knowledge model [25]. Reproduced with permission from [25], Research in Mathematics Education.

The subdomain Knowledge of Topics (KoT) is defined as a deep, grounded knowledge of mathematical subject matter in terms of the following categories: Phenomenology and Applications, which encompasses knowledge of situations to which the meanings associated with a specific topic can be applied, as well as other applications of the topic;

Definitions, Properties, and Foundations, which comprises the knowledge required to describe or characterise a concept, along with the knowledge of the properties inherent to a mathematical object; Registers of Representation, which concerns knowledge of the ways in which a topic can be represented mathematically (for example, notation and mathematical language), and Procedures, which itself comprises the subcategories of knowledge of procedures, both conventional and alternative (How is something done?), the conditions needed to proceed (When can something be done?), the basics of the procedure (Why is something done this way?), and the characteristics of the resultant mathematical object (Characteristics of the Result).

The subdomain Knowledge of the Structure of Mathematics (KSM) is concerned with how mathematics is internally interconnected, whilst Knowledge of Practices in Mathematics (KPM) consists of knowledge about how mathematics is constructed (ways of proceeding), such as what conditions are necessary to generate definitions, or how to validate and demonstrate. Given that these two subdomains are not the focus of this article, we will not describe them in any further detail here; for a fuller description, readers are referred to Carrillo [25].

The subdomain Knowledge of Features of Learning Mathematics (KFLM) focuses on mathematical content in terms of how it is learnt, distinguishing the following categories: Strengths and Difficulties, concerning elements that might enhance or hinder learning a mathematical topic; Ways of Interacting with Mathematical Content, dealing with the typical strategies students employ to help them learn a particular item; and Interests and Expectations, contemplating students' affective response to the subject matter.

Knowledge of Mathematics Teaching (KMT) is composed of the following categories: Theories of Teaching specific to mathematical content; Material and Virtual Resources as tools for mathematics education; and Strategies, Techniques, Tasks, and Examples for teaching, including knowledge of their potential.

Finally, the subdomain Knowledge of Mathematical Learning Standards (KMLS) encompasses awareness of the specifications and recommendations provided by the curriculum, scientific journals, research groups and/or professional associations. It covers knowledge concerning what needs to be taught and when, what level of understanding is expected of the students, and what sequencing should be applied to the subject matter.

We recognise that teacher knowledge is a complex and dynamic system which cannot be compartmentalised, and we concur with Scheiner [24] when they point out that a mathematics teacher's specialisation does not lie in the sum of the components making up his or her knowledge, but in the combination of various facets of knowledge informing and interacting with each other to form emergent structures. The use of the MTSK as the theoretical framework in this research is intended to facilitate studying teacher knowledge and identifying links between elements of this knowledge (thus helping us to understand how it is integrated), whilst remaining aware that the teacher's content knowledge and pedagogical content knowledge complement each other.

In this respect, the interconnections between two or three MTSK subdomains contribute to our knowledge of how the different facets of the teacher's knowledge fit together. In the same way, they explain how the teacher's mathematical knowledge provides support for their classroom performance and represent a starting point for research into the development of teachers' content knowledge and pedagogical content knowledge [29].

## 3. Research Methods

The aim of this study is to identify the content knowledge and pedagogical content knowledge of a lecturer in linear algebra, and to uncover the interconnections between elements of these knowledge domains, with a view to gaining insights into their practice. It is a qualitative study, designed within the framework of an instrumental case study [30], since we are trying to understand the specialised knowledge of a teacher who teaches the content of matrices, through the observation of his practice and an interview. The research question is: what interconnections between content knowledge and pedagogical content

knowledge can be identified from a university lecturer's classroom practice when dealing with the topic of matrices?

### 3.1. Context and Choice of Subject

The lecturer participating in the study, who for the purposes of this paper will be called Jordy, was a graduate in the Educational Sciences, majoring in Mathematics, and since 2006 had taught mathematics-related courses on propaedeutic (foundation) programmes at a university in Ecuador. In addition, from 2009 to the time of this study, he had also taught linear algebra on an Engineering degree course. He was chosen for his experience in teaching the subject and for his training in education (as opposed to other candidates with purely disciplinary backgrounds). In addition, for his willingness and motivation to collaborate in research, and because we intend to intervene a posteriori with a proposal for the implementation of didactic activities elaborated with the participation of a group of mathematics professors, we believe it necessary to inquire in the first instance about the domains of the specialised mathematical knowledge of the professor.

The linear algebra module was 16 weeks long and comprised two sessions of two hours per week, with around 20 students. The course syllabus started with matrices and determinants—the topic we selected for making observations on the interconnections between the teacher's content knowledge and pedagogical content knowledge—which was the foundation for later topics related to linear algebra, such as vectors in $R^2$, $R^3$, and $R^N$, vector spaces, and linear transformations.

### 3.2. Data Collection and Analysis

Data collection was carried out through non-participating classroom observations (using video recordings) and semi-structured interviews. In order to collate as much information as possible, recordings were made of Jordy's lessons on matrices and determinants over two consecutive academic years, providing a total of 13 sessions.

This and other data form part of a large-scale research project into the knowledge and conceptions of university lecturers in linear algebra [31]. In this paper we illustrate the interconnections between the teacher's content knowledge and pedagogical content knowledge based on the analysis of various episodes (lesson excerpts dealing with a specific topic) from the teaching sessions and interview extracts.

The video recordings were then transcribed and subjected to content analysis [32] in order to identify instances involving the deployment of knowledge corresponding to the categories comprising the MTSK subdomains, always supported by evidence from the teacher's actions and commentaries in class. Two semi-structured interviews were subsequently carried out so as to contribute additional information and to validate some of the researchers' interpretations of the teacher's knowledge.

Our intention throughout was to answer the research question posed by the study through the application of the Mathematics Teacher's Specialised Knowledge framework, given that, as noted above, there are so few studies on university lecturers' knowledge.

## 4. Results

Below we explore the teacher's specialised knowledge by analysing the interconnections between the different facets, in other words, Knowledge of Topics (KoT), Knowledge of Features of Learning Mathematics (KFLM—Strengths and Difficulties), and Knowledge of Mathematics Teaching (KMT—Strategies, Techniques, Tasks, and Examples), as supported by evidence from the classes. The depth of the interconnections thus identified shed light on how this teacher carries out his work. In the transcriptions of episodes illustrating the analyses below, J refers to Jordy (the teacher) and S to any student.

One episode which provides a good illustration of Jordy's teaching occurs in a class introducing the topic of how to multiply matrices:

J:　　To add matrices we need one condition. What is it?
S:　　They must have the same dimension.

J:　To multiply them we also need a condition. If we've got the matrix A = $\begin{bmatrix} 2 & 3 & 1 \\ 4 & -5 & 0 \end{bmatrix}$ what are the dimensions of this matrix?

S:　Two rows, three columns.

J:　The dimension of this matrix is 2 × 3. To be able to multiply two matrices, we need the number of columns in the first matrix to be the same as the number of rows in the second. If A is like that, B must necessarily have three rows, the number of columns doesn't matter. Let's suppose that matrix B is a column matrix, B = $\begin{bmatrix} 1 \\ -3 \\ 5 \end{bmatrix}$. We can multiply with this, the condition is that it has three rows. What are the dimensions of this matrix?

S:　Three by one.

J:　Three by one. If it does not fulfil this condition, then multiplication is not possible. The matrix B can also be, for example, B = $\begin{bmatrix} 2 & 3 \\ -1 & 5 \\ 7 & 0 \end{bmatrix}$. What are its dimensions?

S:　Three by two.

J:　Yes, so if you want to multiply A × B, you can because the number of columns in matrix A coincides with the number of rows in matrix B. And it can be another matrix with any number of columns, for example B = $\begin{bmatrix} -1 & 0 & 3 & 4 \\ 5 & 2 & 3 & -1 \\ 2 & -2 & 5 & 0 \end{bmatrix}$. (Extract 1.)

It can be seen how Jordy creates a context (employing examples and questions directed at the students) and demonstrates his knowledge of the conditions for carrying out the addition and multiplication of matrices (KoT—Procedures, when can something be done?). He highlights the conditions that the matrices must fulfil, giving three possible examples of the second matrix, B, given the dimensions of the first. The set of examples illustrate the feature which matrix B is required to meet, but significantly does not include a 3 × 3 square matrix. His actions in this sequence provide evidence of his KMT (Strategies, Techniques, Tasks, and Examples).

Following this episode, Jordy explains how to multiply two matrices, showing his knowledge of the algorithm for multiplying matrices and the characteristics of the results. He then gives the students an exercise in multiplying matrices, with the following warning:

J:　Before you do the multiplication go back over what we did. The first factors must be the numbers in the rows of the first matrix and the second factors those in the columns of the other matrix. The position of the factors is so that you make fewer mistakes. It is always a good idea to define the dimensions of the matrices to avoid any kind of error in the product. (Extract 2).

Jordy encourages the students to look closely at the dimensions of the matrices in order to avoid errors in the multiplication, and stresses how the procedure is carried out. When asked about this in the subsequent interview, he explained his view of the students' potential errors:

J:　The first error they can be prone to commit is that they think you must multiply number by number according to their position. So, at least in this case when we're multiplying matrices, I tend to emphasise the dimensions. If I've got two square matrices [of order 2], the guys can get a 2 × 2 matrix as a result, which is logical. But on the other hand, they might go about it the same way as in addition and multiply the elements in each matrix with those in the corresponding position in the other one. (Extract 3).

Jordy is aware of the typical mistakes that can occur when multiplying matrices, such as not taking into consideration their dimensions and making a false generalisation from the addition of matrices (the latter explaining why he underlines in Extract 2 that multiplication is carried out according to "first matrix rows" by "second matrix columns")

(KFLM—Strengths and Difficulties). His KoT with respect to Procedures has parallels with his KFLM with respect to Strengths and Difficulties. Jordy's awareness of the importance of the dimensions when multiplying matrices is linked to his awareness of the mistakes that students often make by carrying out the operation without defining the dimensions. His knowledge of the algorithm for multiplying matrices, along with the great care he takes to explain the procedure step by step, forms an interconnection with his knowledge of the kind of errors that can result from wrongly applying the procedure for addition to multiplication.

In addition, the examples that Jordy selects for explaining the significance of the dimensions in multiplying matrices are fundamental to his purpose, as can be seen in the following extract:

J:   So, you've got (3 × 5) (4 × 3). Is it defined? Why?
S:   No, because the columns and rows don't match in size.
J:   [ . . . ] no, the product is undefined.
S:   So, when we're multiplying matrices the order they're in does matter.
J:   What do we mean by that? How can we put it? There we did A(2 × 3) × B(3 × 1), but if we switch the order to B(3 × 1) × A(2 × 3) then in this case we can't do the multiplication. Why's that?
S:   It's not commutative.
J:   Correct, the commutative property does not hold for the multiplication of matrices (A × B ≠ B × A). It's generally not commutative, firstly because of the dimensions, but also even if it could be with square matrices, these are rarely commutative. (Extract 4.)

Jordy's awareness that the product of matrices is non-commutative shows a degree of depth in that he not only makes reference to the property but is also careful to circumscribe the knowledge with a consideration of different possibilities, affirming that even in the case of square matrices the product is not always commutative. Furthermore, when he focuses on which properties fulfil these operations, he is rigorous in his consideration of cases and possible exceptions (thus taking care to ensure the properties are well-founded). The knowledge that this episode demonstrates (KoT—Definitions, Properties, and Foundations) is linked to Jordy's KFLM regarding the errors which students can make as a result of generalising the commutative property displayed by two real numbers to the multiplication of matrices. This accounts for his preference for rectangular and non-square matrices to deal with operations on matrices, and for his affirmation that even in the case of square matrices, the product is not always commutative. It also attests to his knowledge of examples for teaching (KMT—Strategies, Techniques, Tasks, and Examples), which can be seen when he explains the multiplication algorithm using four rectangular matrices as examples in Extract 1.

The interconnections of Jordy's knowledge follow a pattern which repeats over the course of his classes. Thus, for example, when he deals with the topic of stepped matrices, he again does so through the use of three examples, as can be seen in the following episode:

Now I want you to look over here at the board, we've got the matrices $A = \begin{bmatrix} 1 & 2 & 3 & 4 \\ 0 & 1 & 2 & 1 \\ 0 & 0 & 0 & 3 \end{bmatrix}$,

$B = \begin{bmatrix} 1 & 5 & 0 & 2 \\ 0 & 0 & -3 & 4 \\ 0 & 0 & 0 & 0 \end{bmatrix}$, and $C = \begin{bmatrix} 0 & 1 & 3 & 8 \\ 0 & 0 & 0 & 2 \\ 0 & 0 & 0 & 0 \end{bmatrix}$. What's happening in the first matrix, can you see anything interesting? First, they're not square. What's special about them?

S:   A few zeros.
J:   And what are the zeros doing?
S:   They're increasing.
J:   The zeros are increasing as you go down the rows. The important thing is that there are zeros before a nonzero element in the row. This kind of matrix is called a stepped matrix, it looks as if it has steps, and as you go down the rows the zeros increase until you get a nonzero number or until the whole row is zeros. The first nonzero

element in each row is known as the distinguished element of the row. There can only be one distinguished element per row, it doesn't matter if after that there are zeros. (Extract 5).

Jordy points out to the students the salient features of the definition of a stepped matrix (KoT—Definitions, Properties, and Foundations) and demonstrates his knowledge of registers (KoT—Registers of Representation) with respect to algebra and matrices [33] in representing each stepped matrix and of diagrammatic schemes [34] with respect to the clarifying elements he draws on the matrices, such as the overlays marking the stepped format and the highlighted numbers indicating the distinguished elements for each row. In one of the subsequent interviews Jordy explained his intention to write these examples of stepped matrices:

J:     My intention was to give them three different matrices, each one stepped in its own way so that they would realise that there are different kinds of stepped matrices and see the essential thing about them because it's easy to make mistakes, thinking they're stepped when they're not. (Extract 6).

The interview extract provides further elucidation of Jordy's knowledge of errors (KFLM—Strengths and Difficulties). He builds a definition of a stepped matrix for the students (KoT—Definitions, Properties, and Foundations), providing three examples which focus on the essential features of this kind of matrix. He creates a link between his KoT, which he draws on to tell the students the defining properties (" . . . it looks as if it has steps, and as you go down the rows the zeros increase until you get a nonzero number or until the whole row is zeros"), and his KFLM, when he alerts the students to potential errors arising from ancillary properties ("There can only be one distinguished element per row, it does not matter if after that there are zeros"). There is evidence, too, of a connection with his KMT regarding the examples he uses to introduce the topic of stepped matrices, which, as when he dealt with operations with matrices, are selected to bring to the fore important features of the content and illustrate elements which can be the cause of error.

During our analysis of Jordy's practice, we have been able to establish a recurrent pattern of connections between his KoT, KFLM, and KMT (Figure 2). Thus, whenever he introduces a new topic (which naturally involves his KoT regarding Procedures, Definitions, Properties, and Foundations), he draws support from a variety of examples (KMT) through which he is able to highlight salient features of the topic and direct students' attention towards potential errors (KFLM) concerning the content and mathematical notation (KoT—Registers of Representation). Jordy frequently brings his awareness of potential error to bear on his classes in conjunction with carefully selected examples, which he deploys as an illuminating educational resource.

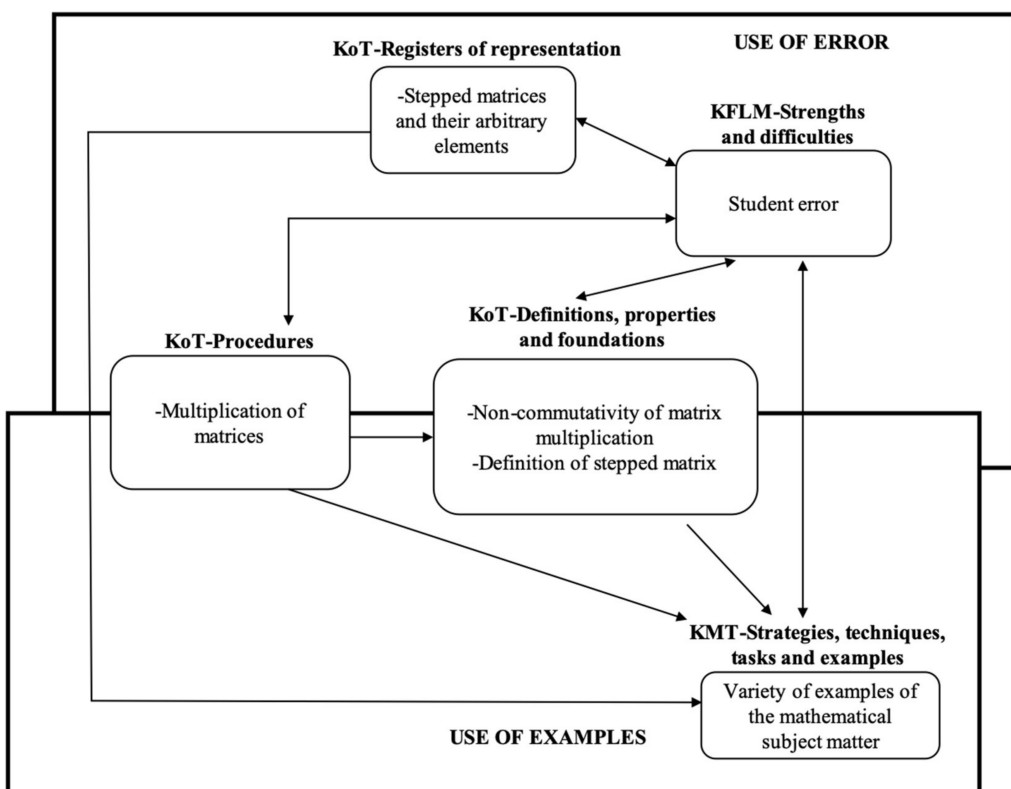

**Figure 2.** Main connections between elements of Jordy's specialised knowledge of matrices.

## 5. Discussion and Conclusions

The interconnections identified through the analysis of Jordy's knowledge essentially reflect knowledge of topics (KoT) linked to knowledge of student error (KFLM) and the use of examples (KMT), which are deployed for explaining mathematical subject matter. At the same time, these interconnections help us to understand aspects of Jordy's teaching as they provide the background to his repeated warnings of potential errors the students could fall into. His knowledge of errors is formulated in terms of how they can be used to improve the learning process, as he has been able to analyse potential errors regarding the mathematical content which the students might commit [22].

This knowledge of the typical errors associated with the subject matter in question, acquired over his years as a teacher, has been incorporated into his explanations with the aim of preventing his students from inadvertently committing them. It is thus deployed as a teaching strategy and is consonant with the discourse modes proposed by Jaworski [14]. Hence, when Jordy is discussing the topic itself, he employs the expository mode, whilst when he is focussing on strategies for improving student understanding, he utilises the didactic mode. Some of the errors specific to this topic anticipated by Jordy coincide with those identified by Sosa [11] regarding secondary teachers' knowledge, including the product of a matrix with scalar and multiplying matrices. From among the indicators by which these authors describe teachers' Knowledge of the Features of Learning Mathematics, we can identify in Jordy, above all, knowledge regarding student errors and difficulties (student needs and difficulties with respect to the subject matter, misunderstandings arising from making incorrect connections with previous items, and errors from contravening mathematical conventions).

The examples that Jordy deploys provide support for his exposition of the subject matter and reflect his knowledge of mathematics teaching. Consonant with Mali [17], these examples can be categorised as generic, serving to illustrate essential features of a mathematical topic (such as distinguishing the algorithm for matrix addition from that for matrix multiplication, underlining the non-commutativity of matrix multiplication, and

identifying the characteristics of stepped matrices). It can also be noted that this knowledge and the way in which it is deployed mediates student understanding, enables Jordy to warn them of potential errors, and constitutes a teaching pattern that is repeated over various sessions.

Continuing with the focus on examples, according to the taxonomy devised by Figueiredo [18], those chosen by Jordy pertain to the category of denominated characteristics, intended to lead students to a deeper understanding of the different aspects of the concept in question by illuminating peculiarities (the non-commutativity of matrix multiplication, and the characteristics of stepped matrices), and focusing on difficulties (the position of the zeros and the distinguished elements of the rows in a stepped matrix), the presentation of which is dependent upon the lecturers' forethought, experience, and originality. The examples also have the attributes of transparency and generalisability [16], that is, they embody the appropriate features with which to illustrate concepts and procedures, and they highlight the essential features of the topic in question in contradistinction to those that are arbitrary and changeable.

Likewise, they meet the criterion of variation [19], insofar as they foreground aspects of the topic which can vary, such as the case of the multiplication of matrices, in which Jordy employs four rectangular matrices (excluding squares) so that the students understand the importance of establishing the dimension of each matrix in order to carry out the operation. He demonstrates his awareness of the critical aspects of the topic in question and of how these can be transmitted through a set of examples selected to focus on them [20]. In terms of the parameters of variation theory, these examples can be classified as a fusion [21], foregrounding what varies and what remains the same in a mathematical object. In this instance, the examples of stepped matrices illustrate that there can only be one distinguished element in each row, which can take different positions, or that all the elements in a row can be zeros.

With respect to content knowledge, our analysis identified Jordy's KoT; furthermore, we can say that, according to the descriptors of the epistemological dimension of the model developed by Artigue [6], Jordy regards linear algebra as a subject requiring knowledge of modes of representation. According to this model, Jordy demonstrates knowledge of school algebra, knowledge of the learning process with respect to student difficulties with the topic of matrices (the cognitive dimension), and knowledge of teaching through the use of examples (the didactic dimension).

This study contributes to our understanding of the knowledge deployed by a university lecturer in linear algebra, and the interconnections between different elements within this topic which it has highlighted can shed light on the forms of training designed for this level. For example, epistemological reflection on the different kinds of difficulties related to learning certain content could encourage lecturers to explore aspects of their pedagogical content knowledge. Such aspects might include the use of examples and tasks for helping students overcome areas of difficulty, teaching resources, theoretical approaches to the teaching–learning binomial, and interconnections between geometric, algebraic, and abstract expression [6] of linear algebra. This largely concurs with Speer [15], who argues that the content knowledge put to use in class enables teachers to learn from their practice and can thus be transformed into pedagogical content knowledge.

In a similar fashion, the interconnections between KoT, KMT, and KFLM revealed in the study shed light on how different facets of teachers' knowledge fit together and provides an account for how mathematical knowledge supports classroom actions reflecting the teacher's pedagogical content knowledge [29].

On the other hand, the study of the relationships between content knowledge and pedagogical content knowledge of mathematics teachers provides an overview with which to identify teaching practices and in turn establish relationships with the behaviour of students. For example, as stated by Blazar [35], "teachers who are effective at improving test scores often are not equally effective at improving students' attitudes and behaviours" (p. 146). The author states that certain dimensions of teaching practices predict

students' attitudes and behaviours ("teaching effects"). In their study they found that upper-elementary teachers have a large effect on measures of students' behaviour in math class (including the happiness), concluding that student attitudes are predicted by teaching practices closest to these measures.

Based on the study of the specialised knowledge of a mathematics university lecturer, and its importance for reflecting and establishing teacher training as well as updating programmes, educational technologies could be integrated. Androniceanu [36] indicates that these technologies are the result of the evolution of educational methods and new information and communication technologies. The author refers to learning with lasting effect when selecting educational technologies and argues that the integration of such technologies into training education requires teachers to develop students' thinking.

From another perspective, the application of the MTSK model to Jordy's practice brought to light various interconnections in his knowledge, thus validating the utility of the framework for studying lecturers' specialised knowledge.

It should be mentioned, however, that the decisions taken in regard to data collection might have limited the scope of the results. The class observations were non-intrusive by design, and although follow-up interviews were conducted in order to support the video evidence of Jordy's knowledge and to validate our interpretations as researchers, we feel that the analysis of his specialised knowledge could have been enhanced had we suggested trying out activities and problems in class which promoted a greater degree of interaction between teacher and students.

This study has sought to explain interrelations between different subdomains of a teacher's knowledge, including Knowledge of Topics, Knowledge of the Features of Learning Mathematics (with respect to errors and difficulties), and Knowledge of Mathematics Teaching (with respect to the use of examples). This study could be a starting point for subsequent studies into teachers' knowledge regarding interconnections between mathematical structures (KSM), ways of carrying out mathematical procedures (KPM), and other categories of knowledge within the subdomains of mathematics teaching (KMT) as well as features of the students' learning (KFLM), which were not found in our study. Furthermore, our findings in this study lead us to consider how the MTSK model might be implemented in professional development courses oriented towards university lecturers, a question demanding considerable reflection before we can enter the particulars of future programmes.

**Author Contributions:** Conceptualization, D.V.-M., N.C.-R. and D.E.-Á.; methodology, D.V.-M., N.C.-R. and D.E.-Á.; formal analysis, D.V.-M. and N.C.-R.; validation, D.V.-M., N.C.-R. and D.E.-Á.; writing-original draft preparation, D.V.-M., N.C.-R. and D.E.-Á.; funding acquisition, N.C.-R. All authors have read and agreed to the published version of the manuscript.

**Funding:** This research was financed by the COIDESO research centre at the University of Huelva, the Ministry of Science, Innovation and Universities of the Government of Spain (project: RTI2018-096547-B-I00), and the research group DESYM (HUM-168).

**Institutional Review Board Statement:** Not applicable.

**Informed Consent Statement:** Informed consent was obtained from all subjects involved in the study.

**Data Availability Statement:** Not applicable.

**Conflicts of Interest:** The authors declare no conflict of interest.

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
