# Peer review of "Interconnections between Content Knowledge and Pedagogical Content Knowledge of a University Lecturer in Linear Algebra"

_mathematics, doi:10.3390/math9202542_

Round 1

Reviewer 1 Report

The purpose of this article is to explore the interrelationships between the different kinds of knowledge brought into play by a university lecturer in the course of his practice, and, specifically, the authors wish to raise the question of what elements of content knowledge and pedagogical content knowledge relating to linear algebra underpin this practice. See lines 41-48.  

The authors have divided information into two parts:

  • the first dealing with the background to studying mathematics teacher knowledge at both secondary and university levels,
  • the second describing the theoretical model applied to this study. See lines 64-68.

The authors have shortly before explained to be focused in measuring the lecturer’s knowledge in their chosen case study. Their target is to identify the presence or absence of essential elements required for teaching mathematics, with an interest in understanding the knowledge which the lecturer brings into play in the classroom, and the interrelationships between the various components which comprise it. See lines 49-57.

The focus in Learning Theory is generally modeled on students’ difficulties, while the authors shift it on the knowledge deployed by lecturers in linear algebra. See lines 55-56.

From the empirical side of view the authors have found in this case study that the prospective teachers regarded algebra as a domain restricted to the field of algebraic theory (the epistemological dimension), in which algebraic techniques were limited to the syntactic part as a set of rules. This means this teacher committed himself to move between different semiotic registers and languages, such as the geometric, the algebraic and the abstract. See lines 84-92.

At lines 101-150, the authors enlist other different ways to provide mathematics teaching at university level, focusing instead for instance on (i) the use of examples, (ii) the use of variation, and (iii) the use of error to improve the learning process.

Paragraph 2.2. The Mathematics Teacher’s Specialised Knowledge is a description of the model with a figurative representation in Figure 1. at line 175.

Mathematics Teacher’s Specialised Knowledge (MTSK) involves (i) Knowledge of Topics (KoT), (ii) Knowledge of the Structure of Mathematics (KSM), (iii) Knowledge of Practices in Mathematics (KPM), (iv) Knowledge of Features of Learning Mathematics (KFLM), (v) Knowledge of Mathematics Teaching (KMT), (vi) Knowledge of Mathematical Learning Standards (KMLS). The authors describe in detail the epistemological interconnections into the sophisticated net of the teaching of mathematics at lines 169-226.

At paragraph 3. Research Methods, the authors enter the topic of matrices. They inquired the methods used by their chosen case study—called Jordy, who is a graduate in the Educational Sciences majoring in Mathematics, and since 2006 had taught mathematics-related courses as well as linear algebra on an Engineering degree course—to teach a module of 16 weeks about Matrices and Determinants. See lines 228-248.

The method employed by authors to inquiry Jordy’s teaching of mathematics has been through non-participating classroom observations (using video recordings) and semi-structured interviews over two consecutive academic years for a total of 13 sessions. See lines 250-253.

Content Analysis was used by authors to identify instances involving the deployment of knowledge corresponding to the categories comprising the MTSK subdomains. See lines 259-261.

The authors provide a set of examples (Extracts 1, 2, 3) of Jordy’s teaching at lines 276-335. Thereafter, they comment on the self-feedback this teacher is providing himself from the mistakes that students encounter during the operations because have no correctly defined matrices’ dimensions.

Other examples (Extracts 4, 5, 6) of Jordy’s teaching are provided at lines 353-371, and lines 392-423.

An interesting representation of the main connections between elements of Jordy’s specialized knowledge of matrices, is provided in Figure 2. at line 447.

Paragraph 5. Discussion and Conclusions is a nice summary of the Jordy’s ability to switch between the cognitive dimension and the didactic dimension in teaching mathematics.

REVIEWER’S COMMENT:

  • Please add (i) “Learning Theory”, and (ii) “Content Analysis” to your keywords below the abstract,
  • At line 68 full stop is missing,
  • Please delete one “c” at line 288 in the word “necessarily”; add one “p” at line 289 in the word “supose”; add one “r” at line 296 in the word “matix”,
  • When you enlist other different ways to provide mathematics teaching at university level, I would add an example drawn from focusing students’ achievement on tests—Are there teachers/Learning theories focused on performance of students?

Please see: Blazar, D., & Kraft, M. A. (2017). Teacher and Teaching Effects on Students' Attitudes and Behaviors. Educational evaluation and policy analysis, 39(1), 146–170. https://doi.org/10.3102/0162373716670260,

  • Please rephrase some passages all along your paper—without subverting your chosen structuring—but telling the audience why you have chosen to “embody” the teaching of mathematics in this chosen case study creature, and which target you have about,  
  • Please align reference no. 24 at line 608.

Kind Regards,

Author Response

Thank you for your comments

-All form changes proposed by the reviewer were included in the document. In Discussion and Conclusions, we include information from the following study:
Blazar, D., & Kraft, M. A. (2017). Teacher and Teaching Effects on Students' Attitudes and Behaviors. Educational evaluation and policy analysis, 39(1), 146–170. https://doi.org/10.3102/0162373716670260
We associate the importance of inquiring about the teacher’s teaching practices and their impact on the prediction of students' attitudes and behavior.

-In Research Methods we explain why the case study was selected.

Reviewer 2 Report

It is known that the results of scientific research are usually capitalized in two ways: by immediate practical application or by communication to the scientific community in the field. The authors chose to disseminate the results of their research in the form of an article trying to contribute to the development of knowledge in the field. The fact could be commendable, supported by the importance of the scientific journal in which they chose to disseminate. 

The title of the scientific paper is usually precise, clear and briefly defined, meant to draw the reader's attention to the most important and new idea. A reformulation of the title in this sense would be welcome!

Current research brings more to the lesson plan or methodological guide. A rethinking of the whole approach aimed at contributing to the knowledge in the field would also be welcome!

Viewed from the pedagogical perspective, the dissemination proves a good knowledge of the didactic methods presenting essential aspects of the instructive-educational process.

Thus, the goal of deepening the understanding of the practice of a linear algebra reader by exploring the connections he makes between his knowledge of content and his knowledge of pedagogical content while working on the topic of Matrices is achieved.

From this perspective, I recommend a development of the literature review. The following works can be noted:

Rădulescu, C. V., Burlacu, S., Bodislav, D. A., & Bran, F. (2020). Entrepreneurial Education in the Context of the Imperative Development of Sustainable Business. European Journal of Sustainable Development, 9 (4), 93-93.

Androniceanu, A., & Burlacu, S. (2017). INTEGRATION OF EDUCATIONAL TECHNOLOGIES IN UNIVERSITIES AND STUDENTS'PERCEPTION THEREOF. In The International Scientific Conference eLearning and Software for Education (Vol. 2, p. 26). " Carol I" National Defence University.

In conclusion, I congratulate the authors for the chosen topic (viewed from a didactic perspective) and recommend the publication of the paper after a minor revision.

Author Response

Thank you for your comments

-The reformulation of the title was made seeking precision and clarity: Interconnections between Content Knowledge and Pedagogical Content Knowledge of a University Lecturer in Linear Algebra.

-About the lesson plan or methodological guide: The idea of complementing the study with a lesson plan or methodological guide is interesting and welcome, and precisely this work on the study and discussion of the connections between content knowledge and pedagogical content knowledge of the university lecturer with the MTSK model provides inputs to make a proposal, which we consider will be part of a next article.

In Discussion and Conclusions, we include information from the following study:

Androniceanu, A., & Burlacu, S. (2017). Integration of Educational Technologies in Universities and Students' Perception Thereof. In The International Scientific Conference eLearning and Software for Education (Vol. 2, p. 26). " Carol I" National Defense University.

Reviewer 3 Report

Construction: The paper is well constructed and easy to follow. The writing is accessible and sections are clearly presented. The introduction and background sections provide sufficient information to describe the focus of the paper and to situate the paper in extant literature. The MTSK model is used appropriately to provide structure for the analysis. The results are supported by the analysis of the collected data. Overall the authors show that they are capable of producing a solid journal article.

Content: The data collected was a good fit for the questions asked. Where I have questions is to the impact of the research. While the authors provide a look into how a specific teacher used CK and PK in concert, how these actions affect students learning is the real question. Additionally, including the teacher's thinking about how and why they used PK to enhance CK learning would extend understanding on the relationship between CK and PK.

Overall: The authors have explored an area of mathematics teaching that is only now coming into focus. K-12 education has long been the domain of mathematics education research. College level mathematics teaching and learning presents similar challenges and need to be more deeply investigated. The work of the authors has added to that discussion.

Author Response

Thank you for your comments

Regarding the impact of the research, we do not have enough evidence to speak of a direct impact on student learning but if we could talk about what to recognize the relations between the MK and the PK that could help us to generate training programs or actions with very specific development goals

Round 2

Reviewer 2 Report

The new title realistically suggests the authors' approaches. From a pedagogical point of view, research contributes to the development of knowledge in the field and is an example of good practices that other researchers should pay attention to in their documentations.